# High economic costs of reduced carbon sinks and declining biome stability in Central American forests

Lukas Baumbach [1] ✉, Thomas Hickler[2,3], Rasoul Yousefpour[1,4] & Marc Hanewinkel [1]

Tropical forests represent important supporting pillars for society, supplying global ecosystem services (ES), e.g., as carbon sinks for climate regulation and as crucial habitats for unique biodiversity. However, climate change impacts including implications for the economic value of these services have been rarely explored before. Here, we derive monetary estimates for the effect of climate change on climate regulation and habitat services for the forests of Central America. Our results projected ES declines in 24–62% of the study region with associated economic costs of $51–314 billion/year until 2100. These declines particularly affected montane and dry forests and had strong economic implications for Central America's lower-middle income countries (losses of up to 335% gross domestic product). In addition, economic losses were mostly higher for habitat services than for climate regulation. This highlights the need to expand the focus from mere maximization of $CO_2$ sequestration and avoid false incentives from carbon markets.

The global importance of tropical forests for carbon sequestration and biodiversity conservation has been acknowledged for a long time[1]. While afforestation has become an international standard tool for offsetting carbon emissions, continuing deforestation activities threaten the world's last remaining pristine forests and contribute to accelerating species extinctions[2–5]. On top of these human pressures, changing climatic conditions may affect vegetation growth, composition and structure and trigger biome shifts, rendering habitats unsuitable for species with side effects on ecosystem services (ES) and biodiversity[6,7]. The economic impacts of these changes could be severe for commercial forestry[8], but even more so for the multitude of non-use values that are provided by natural tropical forests[9]. The biogeographic region of Central America, which is home to unique old-growth forests with high levels of biodiversity and high carbon uptake, already experiences climate change impacts and could become a hot

spot in the future[10,11]. In addition, growing land use pressures (cropland expansion, illegal logging activities, etc.) could further reduce the extent of natural rain forests[3,12,13]. The region's complex topography, however, presents difficult conditions for climate impact modelling. The terrain is shaped by mountain ranges that divide the landscape from north to south and influence climatic conditions at a scale that is finer than that of standard climate change scenarios. Previous studies have shown how topography can be a major driver in shaping tropical forest communities and their productivity[14]. Due to the lack of high-resolution future climate scenarios and the high computational effort associated with fine scale analyses, however, large-scale studies on vegetation modelling in our study region so far were carried out at coarse spatial resolutions[15,16]. In contrast, conservation planning and activities such as the delineation of protected areas, establishment of biological corridors or restoration of degraded landscapes usually

[1]Chair of Forestry Economics and Forest Planning, University of Freiburg, Tennenbacherstr. 4, 79106 Freiburg, Germany. [2]Senckenberg Biodiversity and Climate Research Centre (BiK-F), Senckenberganlage 25, 60325 Frankfurt am Main, Germany. [3]Department of Physical Geography, Goethe University Frankfurt, Altenhöferallee 1, 60438 Frankfurt am Main, Germany. [4]Institute of Forestry and Conservation, John H. Daniels Faculty of Architecture, Landscape, and Design, University of Toronto, 33 Willcocks St, Toronto, ON M5S 3B3, Canada. ✉e-mail: lukas.baumbach@ife.uni-freiburg.de

occur at regional to local scales. Furthermore, economic implications of climate change on Central American forest ES up to date remain understudied. Existing analyses have been either performed at local or continental scale and often segregated between ecological and economic modelling of climate change impacts[17–19]. The translation of ecological outcomes into monetary units is however of high importance to provide results for decision-makers in an accessible way, raise awareness for neglected economic implications of ES losses and put their relevance into perspective with other economic figures such as the gross domestic product (GDP). Specifically in view of a growing number of environmental challenges that decision-makers are faced with, the demand for economic estimates of ecosystem service provision remains high, whereas studies providing guiding information are still scarce[20].

Therefore, in this study we aim to bridge some of these research gaps in relation to the following research questions:

1. How may the two ecosystem services climate regulation and habitat be influenced by climate change?
2. Which approximate economic value could be estimated for the provisioning of both services? Which economic costs could arise under climate change?
3. How could these values change when considering national price levels or temporal discounting?
4. Where might we expect hot spots of change which call for action?

We approach these questions by 1) modelling vegetation growth and dynamics with the dynamic global vegetation model (DGVM) LPJ-GUESS at a very fine resolution (1 arc-second) and 2) valuing the provisioning of climate regulation and habitat services economically. Therefore, we estimated net ecosystem exchange in t $CO_2$ (as indicator for climate regulation) and biome stability (as indicator for habitat services) for the time period 1985–2100. Due to the uncertainty of future climate, we ran our simulations for two global climate scenarios with two socio-economic pathways to cover both low and high climate forcing. The modelled indicators were then used to calculate economic ES values by using the social costs of carbon and benefit transfer from existing studies.

For this analysis, we consider habitat as a supporting service and value it through the bundled benefits of other ES that result from it as part of the ES cascade[21,22]. We further explored the sensitivity of these economic estimates towards regionally differing price levels and discounting. Our results are presented as maps showing stable regions and hot spots of changes (Fig. 1), economic estimates per country and in relation to national GDPs (Figs. 2–3) and hot spots of economic costs (Fig. 4). Drawing from the revealed patterns, we finally discuss economic implications for the region's forests. The study findings are of particular importance to assess coupled ecological-economic climate change impacts in Central America for the first time and support the protection of vulnerable and valuable forest ecosystems.

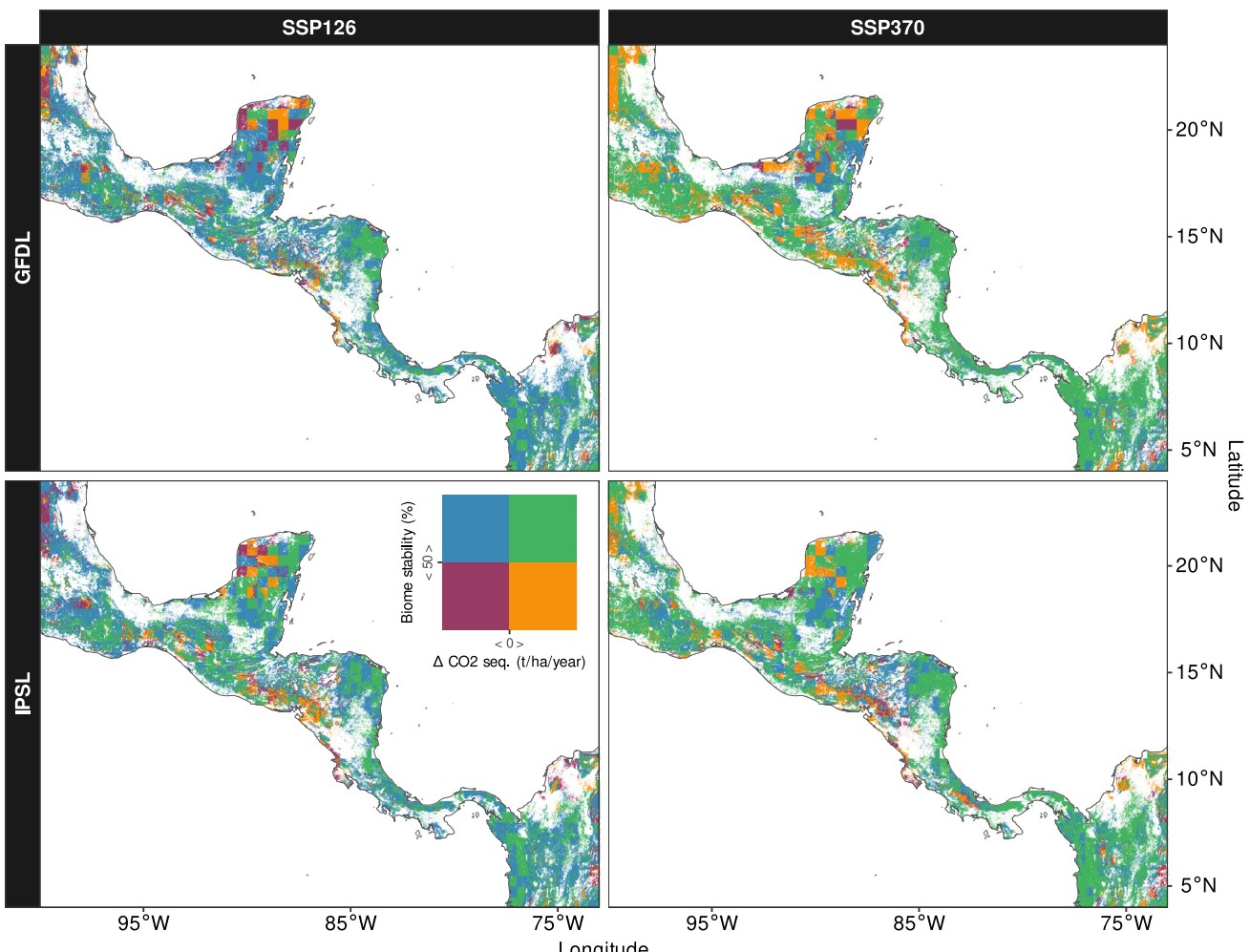

**Fig. 1 | Biome stability and $CO_2$ sequestration.** Bivariate visualization overlaying biome stability and the difference of $CO_2$ sequestration (Δ $CO_2$ seq.) between 2071 and 2100 and the reference period 1985–2014 for all climate scenarios (global climate models: GFDL = GFDL-ESM4, IPSL = IPSL-CM6A-LR; shared socio-economic pathways: SSP126 = SSP1-2.6, SSP370 = SSP3-7.0). The colours correspond to the position within the legend matrix (i.e., magenta: BS < 50%, Δ $CO_2$ seq. < 0; blue: BS > 50%, Δ $CO_2$ seq. <0; orange: BS < 50%, Δ $CO_2$ seq. > 0; green: BS > 50%, Δ $CO_2$ seq. >0).

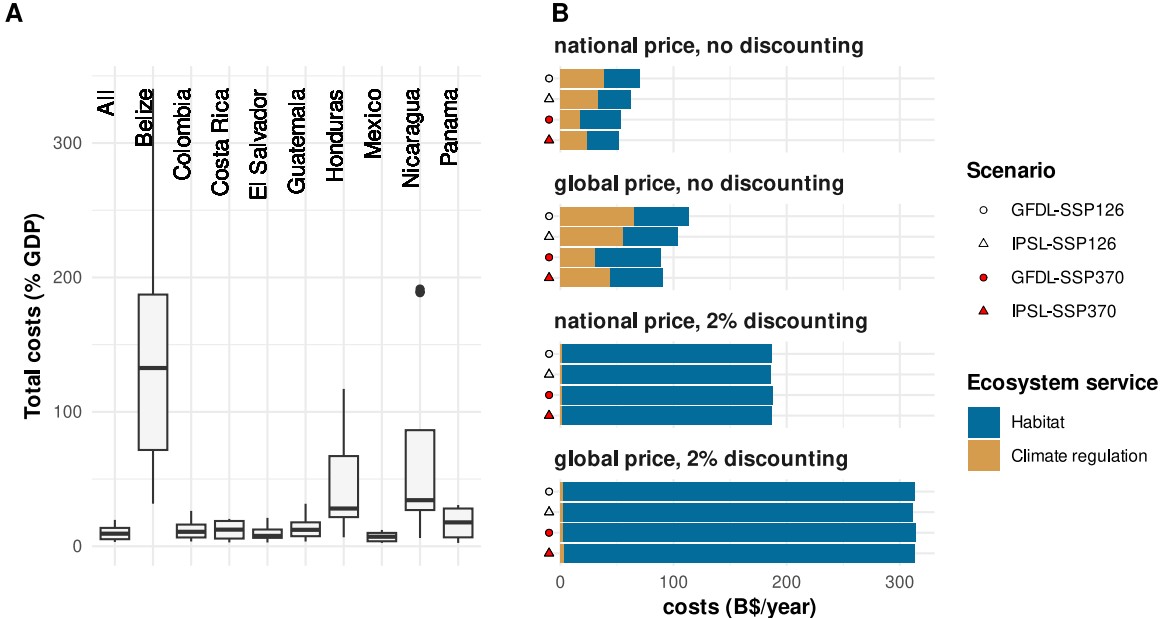

**Fig. 2 | Economic costs of projected ecosystem service declines. A** Boxplots of total costs by country as fraction of their GDP (computed across all price, discounting and climate change scenarios). **B** Costs for ecosystem service declines for the whole study region (billion $/year) for all combinations of price, discounting and climate change scenarios (global climate models: GFDL = GFDL-ESM4, IPSL = IPSL-CM6A-LR; shared socio-economic pathways: SSP126 = SSP1-2.6, SSP370 = SSP3-7.0).

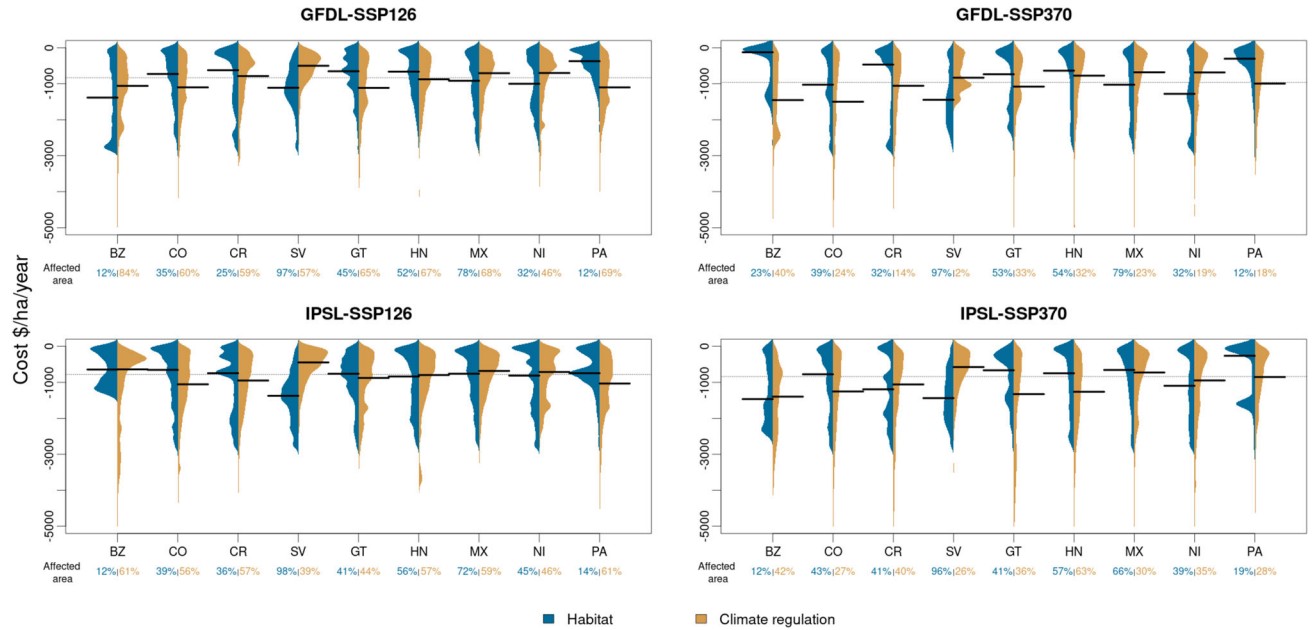

**Fig. 3 | Country-level distribution of ecosystem service losses (global prices, no discounting).** Bean plots showing the distribution and median (bold black lines) of economic costs for declining habitat services (blue color) and decreased climate regulation (dark yellow), separated by country (BZ Belize, CO Colombia, CR Costa Rica, SV El Salvador, GT Guatemala, HN Honduras, MX Mexico, NI Nicaragua, PA Panama). The affected area for each country is relative to the respective national total forest area within the study extent. The thin dotted lines show the median across all countries for each climate scenario (global climate models: GFDL = GFDL-ESM4, IPSL = IPSL-CM6A-LR; shared socio-economic pathways: SSP126 = SSP1-2.6, SSP370 = SSP3-7.0).

## Results

### Spatial analysis of ES indicators

Our simulations covered four climate scenarios, featuring a combination of two global climate models (GCMs), IPSL-CM6A-LR (IPSL) and GFDL-ESM4 (GFDL), and two shared socio-economic pathways, SSP1-2.6 (SSP126) and SSP3-7.0 (SSP370) (for general trends of these scenarios see Fig. S12). To detect long-term changes in our results, we compared the last 30 years of the historical climate data (1985–2014)

with the last 30 years of the climate projections (2071–2100). Mapping the modelled $CO_2$ balance and biome stability revealed regional trends which also varied with climate scenarios. A summary of the directional trends of both variables is shown as a bivariate map in Fig. 1. For simplicity, we applied breaks of 50% biome stability and 0 t/ha/year of $CO_2$ sequestration to the data to be able to distinguish between four major categories of trends (for a continuous presentation of these results, see Fig. S1–2). All scenarios resulted in high biome stability

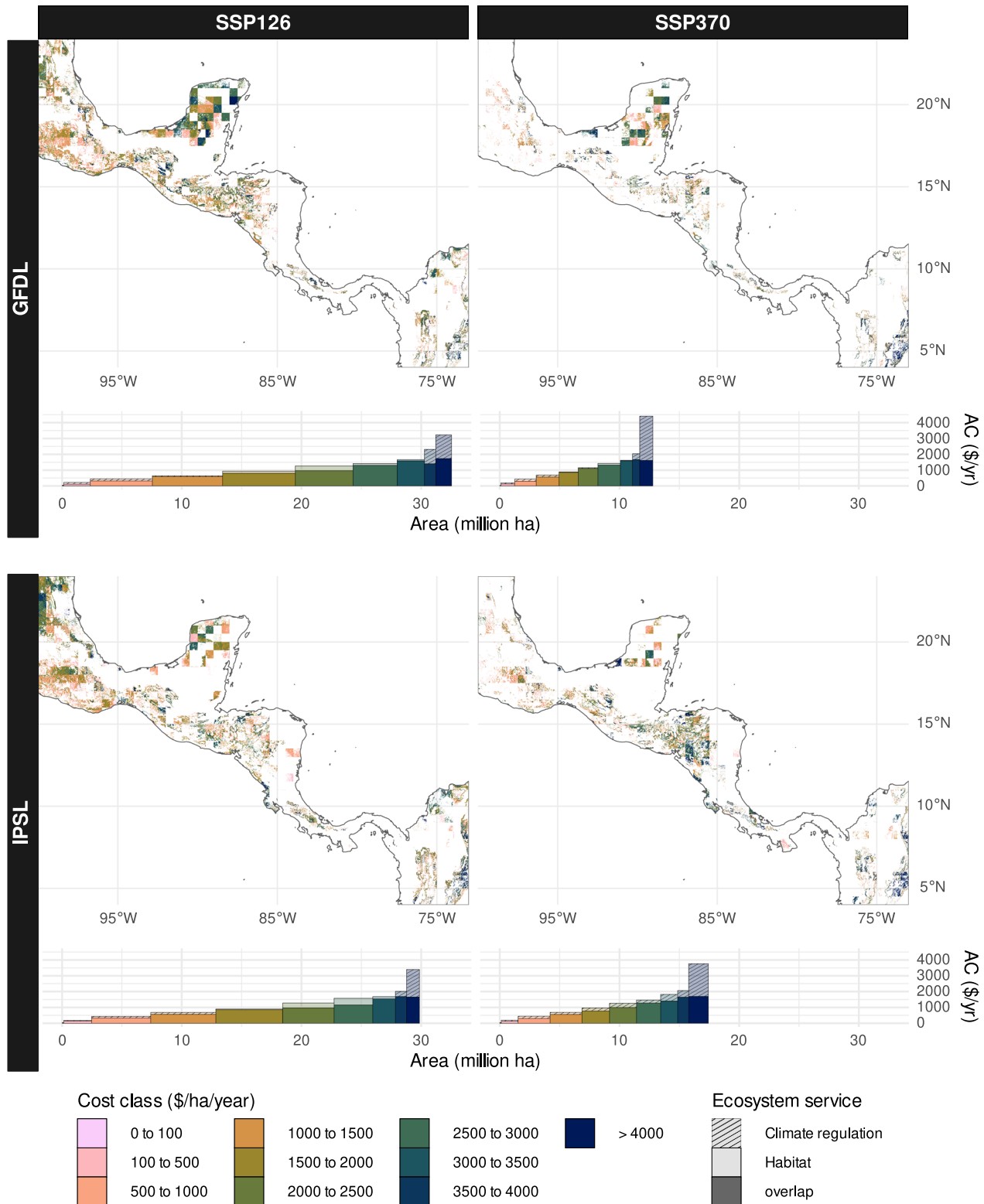

**Fig. 4 | Economic hot spots (global prices, no discounting).** The here presented maps show projected economic losses in areas where both ES declined for each climate scenario (global climate models: GFDL = GFDL-ESM4, IPSL = IPSL-CM6A-LR; shared socio-economic pathways: SSP126 = SSP1-2.6, SSP370 = SSP3-7.0). Cost classes refer to the summed costs of declines in both services. The bar plots show an overlay of the individual ES costs to compare their contribution to overall costs. Bar width shows the area covered by that class, bar height the average cost (AC).

along the Caribbean-facing half of Central America, the southern part of the Yucatan peninsula and parts of Northwest Central America, which largely matches the current extent of the tropical rainforest biome. In the mild (SSP126) scenarios 84.8% and 84.9% (GFDL, IPSL) of the area showed a biome stability above 50%, while these numbers slightly dropped to 79.4% and 84.5% in the SSP370 scenarios (also see Table S1). In terms of $CO_2$ sequestration, a stronger difference between SSPs was found. For the SSP126 scenarios 64.1% and 55.9% of the area

were projected to store less $CO_2$ than during the reference period, while for the SSP370 scenarios lower $CO_2$ sequestration was only projected for 24.1% and 33.4%, respectively. Overall, a consistent trend of decreasing biome stability became apparent in parts of current dry forest biomes along the Pacific coast and in the upper half of the Yucatan peninsula. These areas also featured hot spots, where both $CO_2$ sequestration and biome stability decreased, covering between 4.0% and 8.7% of the study area. Notably, the SSP126 scenarios showed higher decreases of $CO_2$ sequestration, while the SSP370 scenarios resulted in stronger decreases of biome stability. These changes partly showed a non-linear relationship over time, with fluctuating $CO_2$ sequestration and biome stability losses accelerating in the last quarter of the century (Fig. S10–11).

## Economic valuation

Simulated declines in $CO_2$ sequestration and biome stability were translated to economic terms by a) using social costs of carbon estimates for climate regulation and b) transferring ES values from other studies for habitat services. Furthermore, we considered influences of price levels by calculating both a global variant with uniform prices for the whole study area and a nationalized variant where we adjusted prices based on per capita GDP[23]. Also the influence of time preference was tested by either applying a discount rate of 2 % or applying no discounting at all (i.e. a constant ES value over time).

Overall, the economic costs of reduced ES provision ranged between 29 and 313 billion US$/year for habitat and 1–65 billion US$/year for climate regulation (Fig. 2, panel B; for details see Table S2). A major factor for this broad range of results was the application of discounting. For all scenarios, habitat costs calculated with a discount rate of 2% were approximately six times higher than without discounting. Since discounting lead to future ES values for habitat services always being lower than in the historical period, however, costs arose in 100% of the study area irrespective of unchanged biome stability. In contrast, the costs for reduced climate regulation calculated without discounting exceeded the discounted results by a factor of up to 30. The consideration of national inequalities through adjusted prices also showed major effects on the economic outcomes. Generally, cost estimates using uniform global price levels were around 1.6 times higher than estimates with nationally adjusted prices. Regarding the influence of climate, stronger forcing lead to higher habitat costs than the mild SSP126 for the GFDL scenarios, but did not result in higher losses for the IPSL scenarios. While the area with reduced habitat services did not vary significantly between the climate scenarios, the average costs were higher for the SSP370 scenarios. For climate regulation, the opposite trend appeared with higher costs under the mild scenarios due to larger affected areas, although the average costs were again higher for SSP370.

To further break down these trends and cast light from a sociopolitical perspective we summarized the results by country. When put into relation with the size of the economies, total costs from projected ES declines were most marked for Belize, Nicaragua and Honduras where some scenarios resulted in losses of up to 335%, 189 and 115% of the national GDP, respectively (Fig. 2A; for details see Table S3). When applying a uniform global price for ES and without discounting, the habitat costs resembled a right-skewed distribution (i.e. showing a high density of small losses and a longer right tail) for most countries (Fig. 3, for the national scheme see Fig. S3). An exception were El Salvador, where the costs rather followed a normal distribution with a medium to high variance, and Belize, which showed a bimodal distribution under most scenarios. In terms of $CO_2$ costs, variances in general tended to be higher than for habitat costs. Here, a very flat, slightly right-skewed distribution was the most common case with costs ranging up to more than $ 5000/ha/year. Overall, average $CO_2$ costs were higher than average habitat costs in most scenarios for Colombia, Panama, Costa Rica, Honduras and Guatemala, whereas the

opposite was the case for Nicaragua, El Salvador and Mexico and no clear trend emerged for Belize. Under consideration of discounting, the results for habitat costs changed towards normal distributions with a narrow value range around 3100–3200 $/ha/year (Fig. S4, for the national variant see Fig. S5). An exception was El Salvador, which showed a rather uniform distribution of costs. Climate regulation costs again followed a right-skewed distribution for most cases, although at much lower levels with averages around 100 $/ha/year and maxima around 600 $/ha/year.

## Economic hotspots

The above analysis already gives important insights into the sources and distribution of projected costs. From a practical perspective, however, cost mitigation efforts will have to rely on a prioritization of areas due to limited resources. Therefore, in an additional step we filtered areas that showed losses for both services and calculated overall cost per area. Thus identified areas with high costs were termed "economic hotspots". Additionally, we summed up the affected area for different cost categories and also directly compared the contribution of declining habitat services and climate regulation to the total costs. Figure 4 shows the thus uncovered economic hotspots assuming global prices and no discounting (for other variants see Fig. S6–8). For this valuation variant costs for both ES were comparable in most of the area. Only for the highest cost classes the costs for declined climate regulation largely exceeded those for habitat services, yet these only covered small spatial extents. Overall, the most prominent patterns were dictated by the strength of climate forcing. Firstly, the SSP126 scenarios showed altogether larger areas where costs for both ES may arise. Secondly, also the economic hotspots for the highest cost category differed with scenario strength. With global pricing and no discounting, the GFDL-SSP126 scenario showed the highest costs in the Northwest of the study region, particularly over large parts of the Yucatan peninsula, and in montane regions of Honduras, Costa Rica and Colombia. The corresponding SSP370 scenario on the other hand showed less of these areas in the Northwest, but increased costs in the Colombian Andes. The IPSL-SSP126 scenario showed the highest costs along the mountain ranges of the American cordillera from Mexico to Colombia. Finally, the IPSL-SSP370 scenario resulted in a similar pattern, covering less area but showing regionally higher costs in Honduras, Nicaragua and the Colombian Andes. With national prices, however, the distribution of these hot spots changed markedly. There, the highest costs arose in large parts of Mexico, in Costa Rica and Colombia with equal relations between the mild and strong scenarios as in the global scheme. Altogether, the share of the upper cost classes (> 2500 $/ha/year) decreased in all scenarios compared to the global scheme. The application of discounting in combination with global pricing resulted in less distinct but similarly distributed economic hotspots as in the variant without discounting. Finally, the variant using discounting and nationally adjusted prices showed the least informative outcome, since total costs almost exclusively differed between countries and thus merely highlighted costs in the highest income economies.

## Discussion

In this study we estimated potential economic implications of climate change impacts on forest ES. To this end, we successfully applied the process-based model LPJ-GUESS within Central American conditions. The adaptation of climate inputs following the "landforms" approach contributed to an improved representation of temperature, solar radiation and soil depth, yet model inputs for precipitation could not be adjusted in the same manner. Some modelling artefacts hinting at the original 0.5 degree resolution of the input data thus remain. Furthermore, our analysis mainly focused on climate-driven changes of ecosystems. In addition, anthropogenic impacts like land use change may have important implications for Central America's ecosystems in

the future[24]. Alas, an estimation of these effects was beyond the scope of this study.

The here-used valuation techniques (cost-based valuation for climate regulation, benefit transfer for habitat services) are commonly applied in studies on the economic value of ecosystem services[25–27]. However, it should be noted, that a wide range of other valuation techniques is available that could yield different perspectives and results. To cover at least some critical factors influencing economic outcomes, we provided estimates under a range of assumptions (discounting/no discounting, national/global prices). Nevertheless, particularly for the social costs of carbon, large uncertainties exist regarding socio-economic assumptions, damage functions and the choice of an appropriate discount rate, which altogether stimulate the current scientific debate[28].

The general trends of potential biome shifts in this study showed a more conservative picture than the regional simulations by Lyra et al.[29], who used the Inland dynamic vegetation model for Central America and projected biome shifts under RCP4.5 and 8.5 from tropical rain forest to savannas and grasslands for most areas between Mexico and Costa Rica. On the other hand, our results largely agreed with global LPJ-GUESS projections by Huntley et al.[7], who used climatic forcing from the HadCM3B-M2.1aD model under RCP4.5 and 8.5. Altogether, our results suggest that the major part of the current extent of tropical rain forests is at a low risk of biome shifts, even under stronger climate change scenarios. Beyond these areas, however, we found less stability, particularly for the montane forest biome. This trend increased in area and intensity with increasing climate forcing.

Larger $CO_2$ uptake under SSP370 than under SSP126 as projected by our model is largely driven by stronger $CO_2$ fertilization effects, which is consistent with other studies[30–32]. Even though past trends in vegetation greenness and $CO_2$ uptake by terrestrial ecosystems have been partly attributed to $CO_2$ fertilization effects[33,34], the magnitude of these effects on carbon sequestration is highly uncertain[35]. Since the here used LPJ-GUESS model accounted for nitrogen limitation, our simulations might at least be more realistic than earlier vegetation modelling results without nutrient limitation[36]. In contrast to the biome stability trends, mild climate change scenarios resulted in reduced net $CO_2$ uptake in ~60% of the area, while this number was halved in the SSP370 scenarios. Nevertheless, SSP370 resulted in higher extreme values and showed larger regional disparities. In addition to $CO_2$ fertilization effects, areas with decreasing biome stability but increasing $CO_2$ sequestration may also be explained through increasing shares of raingreen PFTs with a high carbon use efficiency. While tropical rain forests have among the highest carbon stocks, tropical seasonally dry ecosystems have been found to achieve comparable net ecosystem productivity[37,38]. On the downside, decreasing biome stability in these areas may pose severe threats to biodiversity conservation, since they include some of the last natural dry forest reserves and already face high pressures from land conversion[39]. Mountain ranges emerged as further hot spots of change with generally decreased biome stability and partly lower $CO_2$ uptake. This may be mainly attributed to increasing temperatures, which could trigger altitudinal shifts of biomes. Overall, considering the high fragmentation of forest landscapes of Central America, implications of biome shifts for conservation may be severe, since species migration is highly restricted and alternative habitat may not be available.

The projected changes in both ES put high economic values at stake over a significant portion of the study area. Even the lowest estimate of 51 billion $/year already represents a considerable fraction of the region's aggregated GDP of 655 billion $[40]. At the same time it needs to be noted, that these costs may accumulate over the course of the century and may show non-linear changes over time. Overall, the application of discounting had the strongest influence on total costs, followed by the choice of price level and only then the magnitude of climate change scenarios. Nevertheless, the spatial distribution of costs was mainly determined by the assumed RCP and to a lesser extent also by the specific GCM. As an important finding, the estimated costs for declining habitat services were always higher than the respective costs for decreased climate regulation for all but one scenario (GFDL-SSP126, no discounting).

Importantly, discounting had opposite effects on the here applied ES values: While the social costs of carbon increase over time, the classical discounting of the net present value for habitat services resulted in lower values for the future than for the historical period. This difference is founded in the underlying logic of the social costs of carbon, which are defined as economic costs of an additional ton of $CO_2$ emitted at a particular point in time. Since economic damages through increasing emissions accumulate over the time, so do the social costs of carbon[41]. Higher discount rates do not change this trend, but reduce the weight of future impacts and thus result in lower social costs of carbon. For continuously supplied ES like habitat on the other hand, discounting results in a stronger preference for short-term gains and reduced valuation of future income in the long term. These inherent differences between both concepts need to be kept in mind when jointly interpreting the discounted results.

From a more social perspective, it needs to be noted that the costs appeared to be unequally distributed over countries and could reinforce existing inequalities. For example, the lower-middle income economies El Salvador and Honduras showed consistently high ES declines, whereas projected losses were always lower for the region's highest-income countries Panama and Costa Rica. Also with regards to GDP, lower-middle income economies such as Belize, Honduras and Nicaragua emerged as most affected countries with losses that may exceed their respective GDP and thus highlight the economic relevance of these services. In accordance with these regional disparities, the choice of price levels also played an important role for overall costs. At the global scale, adverse effects on biodiversity and $CO_2$ sequestration are equally severe, irrespective of where they occur. The economic reality may differ, however, since actual payments are likely to follow national income levels[42]. In the valuation variants with national prices, the areas with the lowest costs due to declining ES quite naturally occurred in the lowest income countries, where mitigation cost per area would thus be lowest. In fact, similar considerations have also been dominating environmental investments in the past, for example in the form of clean development mechanisms or voluntary carbon offsets, which usually take place in developing countries[43,44]. In contrast, a uniform pricing of ES could better reflect the ecological reality and global externalities of ES declines.

The here applied social costs of carbon, though well within the current range of estimates from negative values up to more than 2000 $/tCO_2[45], are still subject to intense scientific debates. A pivotal point in these discussions is their sensitivity towards the applied discount rate, which influences both magnitude (the lower the rate, the higher the costs) and temporal dynamics of the value (the higher the rate, the higher the difference over time). Nevertheless, over the course of the short life span of this concept, estimates have been constantly increasing so far and may continue to do so in the future[28]. Equally, methods for the economic valuation of habitat services are not standardized yet and can vary considerably based on the included proxy ES, valuation techniques, scope or study region. The here applied perspective emphasizes the importance of stable habitat (approached through biome stability) for supporting the provisioning of a bundle of other services along the ecosystem service cascade[21,46]. It needs to be noted, however, that the placement of "habitat" in the context of ecosystem services is still debated because of its complex relations to other services (including classifications as an "ecosystem function" or "intermediate service")[47]. Apart from these conceptual challenges, also valuation methods are in need of better standardization. Particularly for existence or option values economic estimates

may by far exceed six-digit figures (in $/ha/year), whereas other "free" services may be valued at close to zero[48]. To navigate these uncertainties, there is thus both a need for more accurate estimates that consider context dependency and a need for the application of robust metrics when making use of such estimates.

The here projected economic losses, while still being subject to large uncertainties, give a general impression of the magnitude of climate change-related impacts on ecosystem services and associated implications for society. As such, these results may inform incentives that aim at the reduction of greenhouse gas emissions, the conservation of ecosystems or financial compensation for foregone benefits. A common contemporary practice in this regard are payment for ecosystem service programmes. These have received increasing popularity in Central America, ranging from projects via government-funded programs to carbon taxes[49–51]. Also globally the number and financial volume of payments for ecosystem services are growing[52] and there are calls for their extension throughout the tropics (e.g., carbon taxes)[53]. However, the impacts of such payment schemes on the state of nature may differ significantly depending on the targeted ES. A programme purely focused on $CO_2$ sequestration could incentivize forest management regimes that solely aim to optimize productivity. This could lead to more intensive management and promote a shift towards less but more productive tree species or even monocultures[54]. At the same time, the fluctuation of $CO_2$ prices may greatly influence profitability. Over the past decades, more and more markets and tax schemes have been introduced across the globe with a variety of internal mechanisms and gaping price ranges from below $1 up to $119/t $CO_2$[42]. Still, these remain far below most current estimates for the social costs of carbon.

Incentives for habitat conservation on the other hand could mainly promote do-nothing strategies, which could result in trade-offs with climate regulation[55,56]. To make conservation-based PES schemes work, there would also need to be an economically and ecologically worthwhile long-term perspective to prevent species turnover due to frequently changing management or deforestation after the end of a payment cycle. To control tradeoffs and negative side effects, modern market instruments should additionally aim at an integrative approach considering multiple ES[57,58]. As shown in our analysis, the individual costs for climate regulation and habitat conservation could be very high. Mitigation of climate change impacts in the areas shown in Fig. 4 could, however, result in co-benefits for both services as other studies have shown[59,60]. Through careful design of a PES scheme that avoids the above-mentioned pitfalls, synergies between $CO_2$ and habitat conservation may be better harnessed. To this end, differentiations in the guidelines and conditions for payments also need to be made between natural forest ecosystems and degraded landscapes or secondary forests[25]. Any PES scheme, however, hinges on the implementation of accurate and cost-efficient monitoring of the affected ES, which often still limits the application of advanced schemes and needs urgent improvement[61,62].

In summary, our study investigated climate change effects on Central American forest ecosystems and their climate regulation and habitat services. We successfully adapted a dynamic global vegetation model to simulate growth-based indicators for both services at a high spatial resolution. By applying different economic valuation approaches, we derived rough estimates for the development of climate regulation and the provisioning of climatically stable habitats under various climate change scenarios. Our results provide an overview over hot spots of change and potential costs to society in monetary terms and for the first time give an impression at the fine scale needed for local to regional conservation planning. While there is still room for model improvement and opportunities for the application of different valuation techniques, this study takes a first step in linking DGVM simulations with economic value perspectives for Central America.

## Methods

### Study area
The biogeographic region of Central America – as demarcated by the Mesoamerican biodiversity hot spot[63] – covers the landmass between Mexico and Colombia, thus being an ecological bottleneck between the Americas. It comprises a wide variety of ecosystems ranging from montane cloud forests to xeric shrublands condensed in a comparably small geographic area. As a result of topographic complexity and climatic variability, the region is estimated to host at least 17,000 plant species, 1124 bird species, 580 amphibians, 686 reptiles, 509 freshwater fish species and 440 mammals, many of which are endemic[64]. While the overall extent of the hot spot covers 1,130,019 km², the remaining habitat in the early 2000s was only 20% and recently shrank to 14.1%[12,64]. In this study, we focus on the areas hosting tropical forest biomes. We therefore excluded the Northwestern Mexican part of the biodiversity hotspot, but in turn incorporated parts of the neighbouring biogeographic regions of South America for comparison. The southeastern limit was set to the transition area between the Colombian Andes and the Llanos lowlands, which mark a hydrological border and represent difficult edaphic conditions for vegetation modelling due to underlying laterite soils[65]. The thus selected study extent covered the area between 73 and 100 °W and 4–24 °N. Areas strongly influenced by human land use (croplands, urban and mosaics) were excluded from our analysis.

### Environmental forcing data
Bias-corrected climatic inputs were obtained from the ISIMIP3b data set[66–69] at a spatial resolution of 0.5 degrees and at a daily temporal resolution from 1850 to 2100. For nitrogen deposition we used data from Lamarque et al.[70], whereas $CO_2$ data was taken from[71]. For elevation data we used the SRTMGL1 dataset[72] at 1 arc-second resolution, which equals our output resolution. Land cover data for masking areas with human land use was retrieved from the CCI land cover data set v2,1 [73]. A summary of all datasets used in this study is shown in Table S4.

### Vegetation model description
For the purpose of modelling vegetation responses to climate change we used the dynamic global vegetation model LPJ-GUESS (Lund-Potsdam-Jena General Ecosystem Simulator, version 4.0.1.)[74,75]. This climate-sensitive process-based model has been successfully used in a large number of vegetation growth studies at a variety of scales (for an overview see https://web.nateko.lu.se/lpj-guess/resources.html, last accessed 05/01/2022). It is organized as a framework of modules that represent main ecosystem processes and functions including but not limited to: assimilation, respiration, resource allocation/growth, establishment, mortality, disturbance, carbon and nitrogen cycling. The vegetation is modelled as plant functional types (PFTs), which are characterized by functional traits describing physiological attributes and environmental tolerances (Table S5.1, 2). To realize the model at a very fine spatial resolution which accounts for the topographic complexity of our study region, we followed the LPJ-GUESS "landforms" approach introduced by Werner et al.[76]. In a nutshell, we created subversions for each modelling unit based on a topo-climatic classification (for a detailed description and evaluation see Supplementary methods & Fig. S13–14). While our landform modelling approach improved the representation of topographic effects on climate, we did not fully capture the spatial heterogeneity of precipitation. Artefacts caused by the original coarse resolution of 0.5 degrees for precipitation data are thus sometimes still visible in the resulting maps. To improve accuracy, a proper downscaling of precipitation data also with respect to wind fields would be necessary, which alas was beyond the scope of this study.

### Ecosystem service indicators
The estimation of climate regulation through $CO_2$ sequestration was based on net ecosystem exchange (NEE) outputs from LPJ-GUESS.

The raw outputs (kgC/m²) were converted from carbon to $CO_2$ by multiplying with the factor 3.67 (ratio of atomic weights) and to the more practice-oriented notation of t/ha. Subsequently, these values were used to calculate the net $CO_2$ uptake for the reference period 1985–2014 (last 30 years of the historical climate input data) and the future period 2071–2100 (last 30 years of the simulation). Habitat services may on the other hand be approached through other ecosystem services downstream the "ecosystem services cascade"[21,77]. Since the provision of many ecosystem services depends on ecosystem functionality and constancy of supporting services, i.e., habitat, an increasing tendency for biome shifts is also likely to destabilize the provision of ecosystem services and could signal a critical transition as discussed in previous studies by Roche & Campagne (relations between ecosystem stability, functions, and services), Grimm et al. (biome shifts impact ecosystem functioning), and Armsworth & Roughgarden (economic value of ecological stability)[78–80]. Following this, we approximated the reliable provision of habitat through the stability of the underlying biome, i.e., the likelihood of changing vegetation types under changing environmental conditions. To this end, we first classified each stand into one of eight major biomes. The classification mainly followed the biomization scheme by Snell et al.[16]. An exception were the tropical needle-leaved forests (TrNE), which are not parameterized as a seperate PFT in LPJ-GUESS. While in trial runs we tested implementing an additional TrNE class similar to Snell et al.[16], this new PFT showed very little growth, probably due to strong competition with the temperate broadleaved evergreen (TeBE) and C4 grass PFTs. Within the natural forests of Central America the predominant conifers are pines, which co-occur with temperate broadleaves (mostly oaks) and inhabit areas with seasonal drought[81]. With regard to these characteristics, we decided to adjust our classification for this biome and base it on TeBE and precipitation seasonality instead (for the full biomization scheme see Table S6). Finally, the biome stability was calculated by comparing the projected biomes for each year of the future period 2071–2100 to the mean biome of the reference period 1985–2014 (for average maps see Fig. S9). The number of years with unchanged biomes was then divided by the length of the investigated future period (see Eq. 1).

$$stability_i(\%) = \frac{\sum_1^n \left( biome_{reference} \equiv biome_{i,n} \right)}{n} * 100 \qquad (1)$$

where $i$ = stand and $n$ = number of years

## Economic valuation

The constant provisioning of ES forms the foundation for human well-being. In the future, however, some of these benefits may be lost due to a changing climate. Following this, we approached the economic value of the investigated ES from a societal point of view. To this end we estimated net present values for both the historical and future period and calculated their difference as potential costs to society. Since the economic value of costs and benefits may vary considerably depending on the time preference of society, we performed all economic calculations 1) assuming a discount rate of 2% (average from an expert survey by Drupp et al.[82]) and 2) without discounting for comparison. Moreover, with regard to socio-economic inequalities across the study region, the assumed price levels may play an important role for overall economic values. Therefore, we also considered both 1) a global perspective with uniform prices across all countries and 2) a national perspective, for which we adjusted the global price for each country based on per capita GDP in 2020[23] in relation to the global average (see Table S9). For the definition of GDP and respective data we referred to the World Bank database[40]. The classification of countries by income (e.g., lower-middle income economy) used throughout the text also refers to the World Bank classification, which is based upon gross national income per capita and uses the World Bank Atlas method[83].

For the estimation of the economic value of climate regulation, we used estimates for the social costs of carbon by Yang et al.[84]. While new studies with updated estimations for the social costs of carbon are being published every year, we preferred Yang's study since data was available at 5-yearly intervals, for all SSPs 1 through 5 and for a range of damage functions and discount rates. From this dataset we first computed averages across all damage functions for each time interval, SSP and discount rate. To match these estimates with our study setting, we then extracted the values for the year 2015 for the historical period (closest available year) and for the year 2085 for the future period (central value). In the calculations without discounting, we used the values for the year 2085 at 0% discount rate and kept them constant for both the historical and future period.

For the valuation of habitat services, we used a benefit transfer approach. While this approach is not without its caveats[20], it represented the most suitable option for our case due to a lack of primary studies in our study region and a lack of resources to conduct our own analysis. To reduce uncertainties and biases related to specific study settings, we aimed to include a broad database and approximate a central estimate irrespective of the underlying ecosystem. This is also in line with our biodiversity conservation perspective and preferrable to other strategies of unit value transfer[20]. Also, this approach avoids the pitfalls of assigning values by habitat type, which may result in a habitat "ranking" and economic implications that are counterproductive for the conservation of species. Therefore, we searched the Ecosystem Service Valuation Database[48], which is currently the largest compilation of ES value estimates from more than 950 studies. Within this database estimates for our study region were still very scarce, hence we extended the data selection to the tropics (South America, Africa and Asia). We further cleaned the data by removing rows with missing entries and duplicates and selected only values within the 5–95% quantile due to the presence of extreme outliers from 0 to 2.25e + 10 \$/ha/year. While we aimed to include a broad spectrum of ES that are supported through habitat services, we excluded climate regulation (for obvious reasons) and also provisioning services (food, raw materials, water, medicinal resources) from the data set. The latter was done for two reasons: firstly, our simulations focused on natural forests and did not include forest management and secondly, a high and still growing share of forest products is obtained from plantations[85,86], whereas natural forests are mainly valued for regulating, cultural and supporting services[87]. The thus obtained data selection comprised 246 records (for a summary and overview over applied filters see Supplement, Table S7). However, the data showed a strong bias towards tropical rain forest ecosystems. To account for the variety of ecosystems in the study region and arrive at a more general estimate, we grouped all estimates by major biomes (tropical rain forest, tropical dry forest, montane forests, temperate forests and grassland), computed average values for each ecosystem service by biome and summed the results for each biome (similar to the total economic value framework[21]). Finally, we used the average across these biomes and inflation-corrected the value to the year 2015 as baseline for valuing habitat services. For the variants with discounting, we prolongated this baseline value for the historical period (central year 1999, Eq. 2) and discounted it for the future period (central year 2085, Eq. 3).

$$HSV_{1999} = HSV_{2015} * (1+d)^{2015-1999} \qquad (2)$$

$$HSV_{2085} = HSV_{2015} * \frac{1}{1+d^{2085-2015}} \qquad (3)$$

where HSV = habitat service value and d = discount rate

The such obtained economic values were finally multiplied with the values for the ecological indicators for both time periods. An overview over all ecosystem service values used for each respective

valuation variant and (where applicable) standard deviations is given in the Supplementary (Table S8).

## Data availability

ISIMIP3b climatic data is available from the ISIMIP repository at https://data.isimip.org/. Atmospheric $CO_2$ concentrations can be obtained from https://greenhousegases.science.unimelb.edu.au and SRTM DEM data from https://lpdaac.usgs.gov/products/srtmgl1v003/. Economic values for ecosystem services from ESVD are available from https://www.esvd.net/. Estimates for the social costs of carbon are not publicly available, but may be requested from authors of the study of Yang et al.[84]. GDP related data is available from https://data.worldbank.org/. Source data underlying graphs in the main figures is available from the Supplementary Data https://doi.org/10.5281/zenodo.7737544.

## Code availability

Access to the LPJ-GUESS source code is restricted and can be requested from the Department of Physical Geography and Ecosystem Science, Lund University, Sweden. Code including the 'landforms' extension may be requested from the BiK-F research group 'Biogeography and Ecosystem Ecology'.

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

## Acknowledgements

This work has been financially supported by the German Research Foundation (DFG), research grant 416575874, HA5971/2-1. The authors acknowledge support by the state of Baden-Württemberg through bwHPC. We thank the ISIMIP team for making available the ISIMIP input data and Pu Yang for kindly providing extended data on the social costs of carbon. We would further like to thank Matthew Forrest (BiK-F) for guidance around LPJ-GUESS and Christian Werner (KIT/IMK-IFU) for sharing his code and experience regarding the landforms input module.

## Author contributions

All authors jointly conceived the idea. L.B. and T.H. elaborated the modelling framework. L.B., R.Y., and M.H. developed the valuation approach. L.B. performed the analyses and prepared the figures. L.B. prepared the first draft of the manuscript, T.H., R.Y., and M.H. substantially contributed to further revisions.

## Funding

## Competing interests

The authors declare no competing interests.
