## [Peer Review File · Nature Communications]

High economic costs of reduced carbon sinks and declining biome stability in Central American forestsREVIEWER COMMENTS

Reviewer #1 (Remarks to the Author):

The forests of Central America are known for their rich biodiversity and importance for the regional carbon balance. Yet, climate change may put the delivery of the related ecosystem services at risk. The present study aims at contributing to this highly relevant topic, by modelling the consequences of climate change on the forest carbon balance and biome shifts. The relative changes in these two indicators are transformed into monetary units, using a (quasi-)market-price valuation approach. They build on a CO₂ price from the voluntary market and a hypothetical payment for habitat conservation. The authors use this aggregated value of both indicators to identify hot-spots of changes in ecosystem service values.

I highly appreciate the interdisciplinary approach. The application of the renowned LPJ-GUESS model to Central America – which is demanding from a modelling perspective, appears scientifically sound. Analysing trade-offs and synergies between habitat and carbon sequestration outcomes is a topic of urgent need and relevance. Using an ecosystem service valuation approach to do so is certainly a viable method and may help to support decision-makers. However, I have substantial doubts concerning the design and interpretation of the valuation approach. My doubts may also be due to a lack in precision of the described methods. I believe that this is a valuable contribution, which, however, still needs substantial refinement before publication.

My main concerns are described below

1. Clear research question and/or hypotheses: In my opinion the stated goal of “(... modelling vegetation growth and dynamics under climate change at a very fine resolution and evaluating economic implications for the provisioning of CO₂ sequestration and habitat service” is interesting but rather general. It does not clearly reveal an innovative question, method or theory and most likely “undersells” the study’s content. I highly recommend to include one or more specific research question(s) and ideally hypotheses, which the authors should use to structure the results and discussion section. Many interesting aspects, such as the comparison of the national and global valuation approach only appear in the results section, but are not linked to any specific research question and thus the rationale for the comparison is missing. Including more specific research questions could also help to more clearly limit the scope of the study and the interpretation (see my comment below)

2. General logic of the valuation approach: I appreciate the scientific endeavour to couple up-to-date vegetation modelling with ES service assessment. Yet, in this study the additional benefit of economic valuation as compared to an indicator-based trade-off analysis of modelling results (as done in Figure 1) should be clarified, together with a careful reassessment and refinement of the economic terms and valuation method used. First of all the term opportunity cost and the related design and interpretation of the valuation approach should be reconsidered: The TEEB defines opportunity costs as “The benefits foregone by undertaking one activity instead of another”. The authors apply this approach to foregone income from a land-user (ES producer) perspective related to a) income from selling (voluntary) carbon credits and b) receiving PES payments for biodiversity conservation. As far as I understand the study, the costs are not related to a choice of the ES producer between management options, which is the definition of opportunity costs, but rather refers to the effects of an external factor (i.e. climate change affecting vegetation dynamics), which is out of control of the decision-maker. To my understanding, further socio-economic drivers of land-use change beyond changes of natural vegetation are not considered. Therefore, a rewording into “foregone revenues” might already help. An alternative approach would be to use a cost-based approach, such as social costs of carbon, which might be more consistent with the idea of valuing the loss in carbon from a societal (consumer) perspective and the logic of valuing the impacts of climate change.

Concerning the valuation of habitat shifts, the TEEB states that habitat services are

valued through other ES i.e. following the CICES cascade. Therefore, the authors need to describe in more detail how biome stability is related to the average conservation incentive used here. To me the logic of the valuation approach is not convincing. The authors should justify why a change in the natural biome would change the amount of PES received. To my understanding, PES are tight to abstaining from specific management (i.e. timber extraction) and not to a change of plant functional types or biomes. I might have overlooked the relation between the SSP scenarios and LPJ-GUESS simulation but to my understanding changes in hypothetical PES are a poor measure for the marginal value of a change in habitat and related ecosystem functions and services. A more sound alternative approach would be to assess the willingness to pay for the conservation of different PFTs, the value of bioprospecting or option values. I am aware that this is out of scope for this study. Assessing the value of biodiversity is a challenging and highly debated task.

Therefore in sum, I believe that for revealing vulnerability and hot spots of climate-change-induced changes in carbon and biomes, valuation in monetary units is not necessary, especially if the change is simply multiplied by a constant factor. A more detailed information of the analysis given in Figure 1 would be scientifically sound and valuable in itself. I see the advantage of combining both aspects in one value for comparison. However, this approach also holds the downside of a compensatory effect, i.e. lower relative reduction in one service might compensate for a high reduction in the other, while maintaining biodiversity has a high intrinsic value and should not be compensated for by another service. To my opinion a more detailed analysis of impacts of climate change on both indicators, identifying "hot spots" and where exactly differences in the magnitude of impact arise and what the sensitivity exactly is would be more relevant.

The need for transferring ecological indicators into monetary units may be justified if they are to be compared with a productive (commercial) value of provisioning services, i.e. the change in the commercial value of the PFT for the land user. This has not been done as it is probably challenging to derive average commercial wood values for these PFTs. Incorporating such provisioning services and contrasting them with other services could be an interesting addition.

If the authors would like to stick to combining the vegetation model with ecosystem service valuation, I highly recommend to change the perspective from the land owner's opportunity cost perspective to a societal view and value e.g. the resulting social costs of climate-change induced carbon uptake. In terms of habitat stability the above mentioned alternatives could be used. To my understanding the calculated "opportunity costs" are extremely hypothetical and do not reveal the indeed important information of the generated data.

3. The discussion needs a general rewrite and restructuring: I am missing a critical discussion of the limitations of the approach, while the parts on ES are too broad and rather discuss input data than the generated outcome. The main limitation, to my understanding, is that the "costs of climate change" mainly refer to simulated vegetation-dynamics of natural forests, while effects on management and feedbacks on vegetation-cover and ES are not considered. If this is not the case then the authors should better describe the implementation of human impact.

Minor comments:

Abstract

L18: Shouldn't the focus be on deriving monetary estimates of the <effect of climate change on carbon sequestration and habitat>....?

L20: Which are the economic valuation approaches?

L22: I highly recommend not to state that costs of habitat loss are quantified. This strongly oversells the rather simplified and hypothetical approach of using conservation payments. A more careful term should be used

L23: I am not sure whether it is advisable to talk about humid tropical rainforest in Central America as "safe havens" considering the high deforestation rate in Central America.

Introduction

General observation: A large number of references is missing in the reference list (e.g. Huntely et al. 2021, Watson et al. 2018 etc.)

L63-65: Here the authors implicitly state that they bridge the research gap of, so-far mostly segregated ecological and economic modelling (additional comment: here the "modelling of what" remains unclear). However, to my understanding economic modelling is not carried out here, i.e. social drivers, land-use change etc. are not considered. Rather the outcomes of a purely ecological model are valued economically. This limitation of the study should be clearly stated from the beginning.

L73-74: The Relevance of the study should be stated more clearly at the end of the introduction section

Results

L78-80: The selection of the Climate models and the SSPs should be justified. The authors also need to clarify why two global climate models have been used and contrasted. This could, e.g. be incorporated into the research questions if this is a relevant aspect. If not, the effect of further climate models could also be shifted to the appendix

Line 88 and following: When interpreting biome change, the (missing) role of land-use change needs to be clarified.

Table 1 requires additional explanation. Do these values refer to the total area investigated?

Lines 98-100 this effect needs some explanation. Is this because biomes change towards those with higher carbon sequestration?

L99-100: If there is a time dynamic in change in the provision of ES shouldn't the related income be discounted then?

Figure S6 needs more description and a clarification of the fluctuation in time

L106: Include prices for what?

L106: "based on" should be changed into "adjusted by capita"

L109: What is the explanation for the effect of the valuation scheme?

L113: Please clarify "higher loss intensities". To what does intensity refer?

L129-130: I do not understand the logic of these sentences. Being cost-effective for payments for conservation would mean that the payment actually leads to the desired outcome, i.e. no conversion of forests. I do not see how this can be revealed in a study, where any forest area is multiplied by a certain price.

L132: what does equal interval mean?

L126: The term cost overlap needs explanation

L134: In the context of cost-effectiveness the logic of the valuation approach needs to be refined: You implicitly state that there are "synergies" if there are cost overlaps. Usually "synergies" are associated with positive outcomes – at least in an economic interpretation. I believe here you refer to a high correlation of adverse effects of climate change. Synergy would also refer to management options. But this would provide that biome shifts could be halted, which would then inherently also benefit carbon sequestration. In my opinion this is another example where the simulated natural vegetation dynamics caused by climate change is misinterpreted as a land-use activity that can actively be managed.

Fig. 3: I highly recommend not to repeat the legend in the text but instead give more explanation. I had difficulties capturing the logic of the bar diagram below the maps.

Discussion

L168: Very general sentences, should be omitted

L175: Include ...rain forests "in the study region"

L187-188: The sentence was unclear to me

L201-203: I believe comparing a very hypothetical payments for habitat conservation (to be paid by tax payers) with the GDP does not appear scientifically sound. This would be acceptable using e.g. social costs of carbon or any option values of biodiversity but not using hypothetical incentives.

L203: nonlinear in terms of what? Time?

L213: If I understood correctly the value of habitat is not a market price but a hypothetical incentive

L214: I doubt that this study adequately estimates the costs of lost habitat. This should be rephrased

L221-224: Again I have doubts about the logic of "cost efficient". The authors talk about the "efficiency" of a valuation scheme. As far as I understand it should be clarified that the valuation points towards compensating farmers for foregone revenues due to climate-change related vegetation dynamics. However, this is extremely hypothetical given that neither CO2 payments nor forest conservation payments are in place in most of the countries. Furthermore, conservation payments might even become more "cost efficient" in themselves under biome shifts. This is for example if tropical humid forests would turn into tropical dry forests with low commercial value on sites with low suitability for agricultural crops then compensation payments could become even lower in order to maintain forest cover. Thus in sum, I have not understood the political and scientific relevance of the aspect of "cost-efficient" mitigation of the above mentioned "opportunity costs". I would recommend to change the perspective of valuation from the opportunity cost of the land owner or ES producer to a broader societal perspective (e.g. ES consumer), e.g. by using social costs of carbon and e.g. option values and bioprospecting of habitat. From my opinion, cost-efficient compensation could only be calculated by a change in the provisioning services driven by climate change from the forest owner's perspective.

LL226-240: This part is well-written and clear but I feel it is poorly connected to the findings of the study. It could be omitted in exchange of a critical interpretation and discussion of the results. What do these aspects mean for the interpretation of the results? This study does not allow for any connection of payments and management. Thus I think the main message here is that CO2 payments might lead to a shift towards more productive species, meaning that simulation results would then be underestimated?

L253: The authors refer to a valuation scheme or a "compensation" scheme? If the latter is the case this would be disconnected from the findings here

L256-258: I have doubts that this conclusion can be drawn from a study in which management and real opportunity costs of alternative land uses are (to my understanding) not considered. A careful discussion of the limitations of the study and scope of interpretation should be included.

Methodss L276-286: This part could be considerably shortened in favor of a more detailed description of the economic valuation approach and its connection to simulation results which might have driven some of my doubts

L333: By definition of the TEEB Habitat services should be valued through other ES, considering also the CICES cascade. The approach taken here should be discussed in this light

L358: Include "hypothetical" PES

L387: The term "habitat opportunity cost" is new to me and needs to be defined

Reviewer #2 (Remarks to the Author):

A very needed study on the economic impact of climate change in Central America, both due to the region's social vulnerability to climate change, as well as the future impact on a highly biodiverse part of the planet. Furthermore, these social-ecological vulnerability is also reflected in the region's dependence on nature-based economic activities such as ecotourism.

A small detail: CO2 sequestration is an ecosystem function, therefore when you refer to ecosystem services it is more appropriate to call it climate regulation.

I wonder if the altitudinal migration of tree species was considered, and its consequent change in CO2 sequestration.

278. It's Colombia, not Columbia.

The economic valuation has significant problems, especially the methods used to habitat provision.

The value/price paid to landowners in carbon markets and/or PES schemes is independent to the value of the ecosystem service, therefore, the opportunity costs derived from the loss of income from C markets/PES schemes cannot be translated as the economic loss of ecosystem services. Furthermore, to make it more difficult to apply the approach the authors applied, the value paid to landowners is often paid under a bundle approach, including the price of more than one ecosystem service (e.g. PES in Costa Rica.).

Instead of using the carbon market value, a more accurate estimate of the social impact of climate change in the loss of forests would be the Social Cost of Carbon, which is several times higher than \$5/tCO₂.

Using the price from the PPSA of Costa Rica, \$61/ha/year as the value of ES is inadequate, since this value was set from the opportunity cost of a farmer to stop farming, and therefore it is not a value of an ecosystem service.

My greatest concern is how the habitat provision service was valued. As said, the method applied is not the correct one. For this type of service, you have to find which are the species (1 or more, depending on your scope and time and budget to conduct the analysis) that benefits society directly, and therefore the forest by protecting these species (e.g. species of fish in mangroves that are of commercial interest for fisheries or that attract tourists). If primary studies can't be conducted, then I'd recommend the authors using benefit transfer.

Point-by-point response to reviewer's comments

Major points

1) Clarity of research questions

R1: "In my opinion the stated goal of "(... modelling vegetation growth and dynamics under climate change at a very fine resolution and evaluating economic implications for the provisioning of CO2 sequestration and habitat service" is interesting but rather general. It does not clearly reveal an innovative question, method or theory and most likely "undersells" the study's content. I highly recommend to include one or more specific research question(s) and ideally hypotheses, which the authors should use to structure the results and discussion section. Many interesting aspects, such as the comparison of the national and global valuation approach only appear in the results section, but are not linked to any specific research question and thus the rationale for the comparison is missing. Including more specific research questions could also help to more clearly limit the scope of the study and the interpretation [...]"

We added a list of research questions in the introduction to clarify our main motivation and focus of this study (lines 70-78):

"In this study, we aim to bridge some of these research gaps in relation to the following research questions:

- 1. How may the two ecosystem services climate regulation and habitat be influenced by climate change?*
- 2. Which approximate economic value could be estimated for the provisioning of both services? Which economic costs could arise under climate change?*
- 3. How could these values change when considering national price levels or temporal discounting?*
- 4. Where might we expect hot spots of change which call for action?"*

Moreover, we restructured the discussion to better match the key topics of these research questions.

2) Relevance of economic valuation

R1: "I appreciate the scientific endeavour to couple up-to-date vegetation modelling with ES service assessment. Yet, in this study the additional benefit of economic valuation as compared to an indicator-based trade-off analysis of modelling results (as done in Figure 1) should be clarified, together with a careful reassessment and refinement of the economic terms and valuation method used."

R1: "Therefore in sum, I believe that for revealing vulnerability and hot spots of climate-change-induced changes in carbon and biomes, valuation in monetary units is not necessary, especially if the change is simply multiplied by a constant factor. A more detailed information of the analysis given in Figure 1 would be scientifically sound and valuable in itself. I see the advantage of combining both aspects in one value for comparison. How Wagner, G., 2021. Recalculate the social cost of carbon. Nat. Clim. Chang. 11, 293–294. <https://doi.org/10.1038/s41558-021-01018-5>ever, this approach also holds the downside of a compensatory effect, i.e. lower relative reduction in one service might compensate for a high reduction in the other, while maintaining biodiversity has a high intrinsic value and should not be compensated for by another service. To my opinion a more detailed analysis of impacts of climate change on both indicators, identifying "hot spots" and where exactly differences in the magnitude of impact arise and what the sensitivity exactly is would be more relevant.

The need for transferring ecological indicators into monetary units may be justified if they are to be compared with a productive (commercial) value of provisioning services, i.e. the change in the

commercial value of the PFT for the land user. This has not been done as it is probably challenging to derive average commercial wood values for these PFTs. Incorporating such provisioning services and contrasting them with other services could be an interesting addition.”

R2: “A very needed study on the economic impact of climate change in Central America, both due to the region's social vulnerability to climate change, as well as the future impact on a highly biodiverse part of the planet. Furthermore, these social-ecological vulnerability is also reflected in the region's dependence on nature-based economic activities such as ecotourism.”

We understand the concerns that outcomes of the economic valuation may not significantly differ from the trends that already become apparent through the combination of the ecological indicators. However, we are convinced that the connection of the ecological with an economic perspective is still worthwhile, even if simplistic links between the two are assumed. Firstly, as described in the introduction of the paper, both ecology and economy are inseparably linked, yet few scientific studies combine these two worlds. Secondly, economic valuations are of utter importance to translate and communicate particularly the non-marketed value of ecosystems to make them more graspable for decision makers and highlight impacts on economic activities. This is also emphasized by reviewer 2. Lastly, we would like to argue, that differences in income levels between Central American countries and the influence of temporal discounting may yield economic results that are quite different from the patterns of ecological indicators which are independent from socio-economic aspects. Since we did not account for this before, we now also included a valuation variant that considers the abovementioned effect of discounting over time (findings described in main text and new figures in supplement).

The suggested estimation of commercial value of provisioning services would indeed also provide an interesting perspective, particularly in terms of trade-offs with other services. However, the type of management would be very important for the provision of material goods. Since the inclusion of management interventions was out of scope for this study and the need for valuing non-marketed ES is becoming more and more urgent, we decided to stay with non-provisioning services (for details see the point “Economic valuation of habitat” below).

3) Valuation approach / perspective

R1: “First of all the term opportunity cost and the related design and interpretation of the valuation approach should be reconsidered: The TEEB defines opportunity costs as “The benefits foregone by undertaking one activity instead of another”. The authors apply this approach to foregone income from a land-user (ES producer) perspective related to a) income from selling (voluntary) carbon credits and b) receiving PES payments for biodiversity conservation. As far as I understand the study, the costs are not related to a choice of the ES producer between management options, which is the definition of opportunity costs, but rather refers to the effects of an external factor (i.e. climate change affecting vegetation dynamics), which is out of control of the decision-maker.”

R1: “If the authors would like to stick to combining the vegetation model with ecosystem service valuation, I highly recommend to change the perspective from the land owner’s opportunity cost perspective to a societal view and value e.g. the resulting social costs of climate-change induced carbon uptake.”

R2: “The value/price paid to landowners in carbon markets and/or PES schemes is independent to the value of the ecosystem service, therefore, the opportunity costs derived from the loss of income from C markets/PES schemes cannot be translated as the economic loss of ecosystem services.”

The rationale behind our original economic valuation approach was to consider existing financial instruments and thus focus on actually available payments for ecosystem services (PES) for forest

owners. Nevertheless, we agree that the here assumed connection between climate change impacts and PES is not very straightforward. For carbon, only global voluntary markets would be available to forest owners at the moment, whereas regional markets are only hypothetical. For habitat conservation, the existing payment schemes are indeed mainly oriented towards sustainable or no management. As such they only consider opportunity costs due to foregone benefits from business-as-usual management. Additionally, they are often not results-based, which makes them independent from climate change effects.

Following the comments from both reviewers, we thus broadly restructured our valuation approach for both ecosystem services to a societal perspective (see the following two points). Importantly, we now estimate economic values for the historical period and future period separately (net present values), before calculating their difference and thus determining economic costs.

4) Economic valuation of CO₂ sequestration

R1: "An alternative approach would be to use a cost-based approach, such as social costs of carbon, which might be more consistent with the idea of valuing the loss in carbon from a societal (consumer) perspective and the logic of valuing the impacts of climate change."

R2: "Instead of using the carbon market value, a more accurate estimate of the social impact of climate change in the loss of forests would be the Social Cost of Carbon, which is several times higher than \$5/tCO₂."

To estimate the value of carbon sequestration from a societal perspective, we followed the suggestions of both reviewers and re-evaluated our results with the social costs of carbon. Since also national price levels can matter in this respect, we kept our previous separation between global and nationally scaled prices to reflect socio-economic inequalities.

We used estimates from Yang et al. (2018), who estimated the social costs of carbon using a variety of damage models and for a range of discount rates. Importantly, these results were available at 5-yearly intervals and for different discount rates. The exact methodology is described in the paragraph "Economic valuation" in the methods section (lines 487-541), which we extended to clarify our approach.

All results were updated with these new estimates, which exceed the previous range of estimates by far (now being in the range from \$1-65 billion /year compared to previous estimates of \$177-577 million/year).

5) Economic valuation of habitat

R1: "Concerning the valuation of habitat shifts, the TEEB states that habitat services are valued through other ES i.e. following the CICES cascade. Therefore, the authors need to describe in more detail how biome stability is related to the average conservation incentive used here. To me the logic of the valuation approach is not convincing. The authors should justify why a change in the natural biome would change the amount of PES received. To my understanding, PES are tight to abstaining from specific management (i.e. timber extraction) and not to a change of plant functional types or biomes. I might have overlooked the relation between the SSP scenarios and LPJ-GUESS simulation but to my understanding changes in hypothetical PES are a poor measure for the marginal value of a change in habitat and related ecosystem functions and services. A more sound alternative approach would be to assess the willingness to pay for the conservation of different PFTs, the value of bioprospecting or option values. I am aware that this is out of scope for this study. Assessing the value of biodiversity is a challenging and highly debated task."

R2: “Using the price from the PPSA of Costa Rica, \$61/ha/year as the value of ES is inadequate, since this value was set from the opportunity cost of a farmer to stop farming, and therefore it is not a value of an ecosystem service. My greatest concern is how the habitat provision service was valued. As said, the method applied is not the correct one. For this type of service, you have to find which are the species (1 or more, depending on your scope and time and budget to conduct the analysis) that benefits society directly, and therefore the forest by protecting these species (e.g. species of fish in mangroves that are of commercial interest for fisheries or that attract tourists). If primary studies can't be conducted, then I'd recommend the authors using benefit transfer.”

We agree with the concerns of both reviewers. Considering the link between ecosystem state and economic benefits of ecosystem services, we considered multiple alternatives to our current approach including a PFT- or biome-dependent valuation, benefits of non-marketed services and willingness to pay (WTP) for the conservation of (threatened) species. As already remarked by reviewer 1, however, the availability of estimations for ecosystem service values by PFT or biome types is very limited and own analyses are out of scope for this type of study (R1: “*A more sound alternative approach would be to assess the willingness to pay for the conservation of different PFTs, the value of bioprospecting or option values. I am aware that this is out of scope for this study*”). Additionally, the resulting “grading” of PFTs/biomes could lead to false incentives because, for example, the least economically profitable ecosystems may still be of paramount importance for biodiversity conservation (e.g. tropical dry forests). To avoid these pitfalls, we value all biome types equally and instead focus on ecological stability, which we approximate through biome stability. We added an explanation of this rationale in the main text (lines 460-466):

“Since the provision of many ecosystem services depends on ecosystem functionality and constancy of supporting services, i.e. habitat, an increasing tendency for biome shifts is also likely to destabilize the provision of ecosystem services and could signal a critical transition as discussed in previous studies by Roche & Campagne 2017 (relations between ecosystem stability, functions and services) , Grimm et al. 2013 (biome shifts impact ecosystem functioning), Armsworth & Roughgarden 2003 (economic value of ecological stability).”

For a monetary estimation of the potential ecosystem service values (ESV) at stake we followed the reviewers' suggestions of benefit transfer. Therefore, we searched the Ecosystem Service Valuation Database (Foundation for Sustainable Development 2021), which is currently the largest compilation of ESV estimates from more than 950 studies. Since data for our study region was still very scarce, we extended the data selection to the tropics (South America, Africa and Asia). As pointed out by reviewer 1, habitat services are best valued through other services. Therefore, we approximated the value of habitat services similar to the total economic value framework by summing all other services – with a few exceptions. Firstly, we excluded values for provisioning services. These are mainly of importance for managed systems like plantations or semi-natural forests, whereas we focused on natural forests in our simulations and thus did not consider forest management. We also excluded climate regulation, since we already value it separately. In the end, we arrived at an average value of 2800\$/ha/year, which is comparable to global estimates for tropical forest (Costanza et al. 2014). A detailed description of the data selection and cleaning process was added in the methods section (lines 512-536).

Again, all results were updated with these new estimates, which also strongly exceeded previous estimates (from former \$269-712 million/year to now \$28-313 billion/year).

6) Discussion of limitations

R1: “The discussion needs a general rewrite and restructuring: I am missing a critical discussion of the limitations of the approach, while the parts on ES are too broad and rather discuss input data

then the generated outcome. The main limitation, to my understanding, is that the “costs of climate change” mainly refer to simulated vegetation-dynamics of natural forests, while effects on management and feedbacks on vegetation-cover and ES are not considered. If this is not the case then the authors should better describe the implementation of human impact.”

We restructured the discussion section and inserted a paragraph on limitations affecting the interpretation of results at its very beginning (lines 216-233).

“In this study we estimated potential economic implications of climate change impacts on forest ES. To this end, we successfully applied the process-based model LPJ-GUESS within Central American conditions. The adaptation of climate inputs following the “landforms” approach contributed to an improved representation of temperature, solar radiation and soil depth, yet model inputs for precipitation could not be adjusted in the same manner. Some modelling artefacts hinting at the original 0.5 degree resolution of the input data thus remain. Furthermore, our analysis mainly focused on climate-driven changes of ecosystems. In addition, anthropogenic impacts like land use change may have important implications for Central America’s ecosystems in the future (Boit et al., 2016). Alas, an estimation of these effects was beyond the scope of this study.

The here used valuation techniques (cost-based valuation for climate regulation, benefit transfer for habitat services) are commonly applied in studies on the economic value of ecosystem services (e.g., Naime et al. 2020, Sumarga et al. 2015, Martínez-Harms & Balvanera 2012). However, it should be noted, that a wide range of other valuation techniques is available that could yield different perspectives and results. To cover at least some critical factors influencing economic outcomes, we provided estimates under a range of assumptions (discounting/no discounting, national/global prices). Nevertheless, particularly for the social costs of carbon, large uncertainties exist regarding socio-economic assumptions, damage functions and the choice of an appropriate discount rate, which altogether stimulate the current scientific debate (Wagner 2021).”

Additional discussions of methodological choices and uncertainties are given in the methods section.

Other points

CO₂ sequestration as ecosystem service

R2: “A small detail: CO₂ sequestration is an ecosystem function, therefore when you refer to ecosystem services it is more appropriate to call it climate regulation.”

Indeed CO₂ sequestration should be primarily seen as an ecosystem function, since it is an integral biological process and unfolds as benefit to humans only through the intermediate step of affecting greenhouse gas concentration in the atmosphere and thus regulating the global climate. We therefore changed the name of this service to “climate regulation”.

Discounting monetary value

R1: “If there is a time dynamic in change in the provision of ES shouldn’t the related income be discounted then?”

referring to L 99-100: “These changes partly showed a non-linear relationship over time, with biome stability losses accelerating in the last quarter of the century.”

A consideration of discount rates is indeed relevant, particularly for comparing economic values over time. Nevertheless, if and which discount rate is appropriate or ethically justifiable – also regarding inter-generational aspects – is still fiercely debated within the field of climate change economics (Beckerman & Hepburn 2007, Groom et al. 2022, Schröter et al. 2020). To cover this factor in our analysis, we introduced a valuation variant using a discount rate of 2%, which a survey by Drupp et al. (2015) found to be the median preferred discount rate among economists. At the other end of the spectrum, we kept the previous valuation variant without discounting (i.e. constant values over time).

Cost effectiveness

R1: “In the context of cost-effectiveness the logic of the valuation approach needs to be refined: You implicitly state that there are “synergies” if there are cost overlaps. Usually “synergies” are associated with positive outcomes – at least in an economic interpretation. I believe here you refer to a high correlation of adverse effects of climate change. Synergy would also refer to management options. But this would provide that biome shifts could be halted, which would then inherently also benefit carbon sequestration. In my opinion this is another example where the simulated natural vegetation dynamics caused by climate change is misinterpreted as a land-use activity that can actively be managed.”

R1: “The authors talk about the “efficiency” of a valuation scheme. As far as I understand it should be clarified that the valuation points towards compensating farmers for foregone revenues due to climate-change related vegetation dynamics. However, this is extremely hypothetical given that neither CO₂ payments nor forest conservation payments are in place in most of the countries. Furthermore, conservation payments might even become more “cost efficient” in themselves under biome shifts. This is for example if tropical humid forests would turn into tropical dry forests with low commercial value on sites with low suitability for agricultural crops then compensation payments could become even lower in order to maintain forest cover. Thus in sum, I have not understood the political and scientific relevance of the aspect of “cost-efficient” mitigation of the above mentioned “opportunity costs”. I would recommend to change the perspective of valuation from the opportunity cost of the land owner or ES producer to a broader societal perspective (e.g. ES consumer), e.g. by using social costs of carbon and e.g. option values and bioprospecting of habitat. From my opinion, cost-efficient compensation could only be calculated by a change in the provisioning services driven by climate change from the forest owner’s perspective.”

We agree that the here applied term and concept of “cost-effectiveness” may be misleading in this context. Our original intention of this additional step of analysis was to highlight areas, which show high economic costs for both ecosystem services (Fig. 3, maps) and could thus be especially targeted, perhaps through active management. Additionally, the bar plots in Figure 3 were intended to 1) show how the magnitude of costs is distributed over the area (cost classes * area) and 2) compare the contribution of both ecosystem service declines to the total cost. For clarification, we reworded this aspect to “economic hotspots”, refined the explanation of these findings in the main text (lines 172-204) and updated the corresponding Figure 3.

Selection of climate scenarios

R1: “The selection of the Climate models and the SSPs should be justified. The authors also need to clarify why two global climate models have been used and contrasted. This could, e.g. be incorporated into the research questions if this is a relevant aspect. If not, the effect of further climate models could also be shifted to the appendix”

We added a sentence in the introduction and methods section to explain our selection of climate scenarios (lines 83-85): *“Due to the uncertainty of future climate, we ran our simulations for two global climate scenarios with two socio-economic pathways to cover both low and high climate forcing.”*

Furthermore, we added a figure showing average trends of all climate scenarios in the Supplementary (Fig. S12) to present the respective magnitude of climate forcing.

Opposed biome stability and CO₂ sequestration effects

R1: *“this effect needs some explanation. Is this because biomes change towards those with higher carbon sequestration?”*

referring to L98-99: *“Notably, the SSP126 scenarios showed higher decreases of CO₂ sequestration, while the SSP370 scenarios resulted in less stable biomes.”*

These trends may appear counter-intuitive, yet have likely been caused through a CO₂ fertilization effect, which has also been found in other studies (compare lines 246-248). Moreover, net ecosystem productivity may not differ strongly between, for example tropical evergreen PFTs and tropical raingreen PFTs. We added a short discussion of this finding (lines 257-261):

“In addition to CO₂ fertilization effects, areas with decreasing biome stability but increasing CO₂ sequestration may also be explained through increasing shares of raingreen PFTs with a high carbon use efficiency. While tropical rain forests have among the highest carbon stocks, tropical seasonally dry ecosystems have been found to achieve comparable net ecosystem productivity (Keenan & Williams 2019, Mendes et al. 2020).”

Altitudinal shifts

R2: *“I wonder if the altitudinal migration of tree species was considered, and its consequent change in CO₂ sequestration.”*

We accounted for topoclimatic effects in our landforms approach by adapting temperature (through lapse rates), solar radiation (depending on slope and aspect) and soil depth (by topographic position index) based on a DEM at the very high resolution of 1 arc-second. Thus altitudinal effects are already included in our results (except for precipitation, which we did not adapt). Yet, we did not explicitly analyse potential altitudinal migration of tree species. Since LPJ-GUESS runs at the level of plant functional types and does not include interactions between grid cells or stands, it is not very suitable to answer this question. These rather fall into the realm of species distribution modelling, which the authors explored in a previous study for the region.

Missing references

R1: *“General observation: A large number of references is missing in the reference list [...]”*

We are thankful for this note. Apparently the reference managing software used in preparing the manuscript did not automatically update all references. References have been added and revised.

References

- Armsworth, P.R. & Roughgarden, J.E. (2003) The economic value of ecological stability. *PNAS* 100 (12) 7147-7151. <https://doi.org/10.1073/pnas.0832226100>
- Beckerman, W., Hepburn, C. (2007) Ethics of the Discount Rate in the Stern Review on the Economics of Climate Change. *World Economics* 8, 187-210.
- Drupp, M.A., Freeman, M.C., Groom, B., Nesje, F. (2015) Discounting Disentangled: An Expert Survey on the Determinants of the Long-Term Social Discount Rate. Centre for Climate Change Economics and Policy Working Paper 195
- Foundation for Sustainable Development (2021) Ecosystem Services Valuation Database 1.0. Last accessed: 7/26/2022, available from: <https://esvd.net>
- Grimm, N.B., Chapin, F.S., III, Bierwagen, B., Gonzalez, P., Groffman, P.M., Luo, Y., Melton, F., Nadelhoffer, K., Pairis, A., Raymond, P.A., Schimel, J., Williamson, C.E. (2013) The impacts of climate change on ecosystem structure and function. *Frontiers in Ecology and the Environment*, 11: 474-482. <https://doi.org/10.1890/120282>
- Groom, B., Drupp, M.A., Freeman, M.C., Nesje, F. (2022) The Future, Now: A Review of Social Discounting. *Annual Review of Resource Economics* 2022 14:1. <https://doi.org/10.1146/annurev-resource-111920-020721>
- Keenan, T.F., and Williams, C.A. (2018) The terrestrial carbon sink. *Annual Review of Environment and Resources*, 43(1): 219-243. <https://doi.org/10.1146/annurev-environ-102017-030204>
- Martínez-Harms, M.J., Balvanera, P., 2012. Methods for mapping ecosystem service supply: a review. *International Journal of Biodiversity Science, Ecosystem Services & Management*, 8:1-2, 17-25. <https://10.1080/21513732.2012.663792>
- Mendes, K.R., Campos, S., da Silva, L.L., Mutti, P.R., Ferreira, R.R., Medeiros, S.S., Perez-Marin, A.M., Marques, T.V., Ramos, T.M., De Lima Vieira, M.M., Oliveira, C.P., Gonçalves, W.A., Costa, G.B., Antonino, A.C.D., Menezes, R.S.C., Bezerra, B.G., Satnos e Silva, C.M., 2020. Seasonal variation in net ecosystem CO₂ exchange of a Brazilian seasonally dry tropical forest. *Sci Rep* 10, 9454. <https://doi.org/10.1038/s41598-020-66415-w>
- Naime, J., Mora, F., Sánchez-Martínez, Arreola, F., Balvanera, P., 2020. Economic valuation of ecosystem services from secondary tropical forests: trade-offs and implications for policy making. *Forest Ecology and Management*, 473, 118294. <https://doi.org/10.1016/j.foreco.2020.118294>
- Roche, P.K., Campagne, C.S. (2017) From ecosystem integrity to ecosystem condition: a continuity of concepts supporting different aspects of ecosystem sustainability. *Current Opinion in Environmental Sustainability* 29, 63-68. <https://doi.org/10.1016/j.cosust.2017.12.009>
- Schröter, M., Crouzat, E., Hölting, L. et al. (2021) Assumptions in ecosystem service assessments: Increasing transparency for conservation. *Ambio* 50, 289–300. <https://doi.org/10.1007/s13280-020-01379-9>

Sumarga, E., Hein, L., Edens, B., Suwarno, A., 2015. Mapping monetary values of ecosystem services in support of developing ecosystem accounts. *Ecosystem Services*, 12, 71-83.
<https://doi.org/10.1016/j.ecoser.2015.02.009>

Wagner, G., 2021. Recalculate the social cost of carbon. *Nat. Clim. Chang.* 11, 293–294.
<https://doi.org/10.1038/s41558-021-01018-5>

Yang, P., Yao, Y-F., Mi, Z., Cao, Y-F., Liao, H., Yu, B-Y., Liang, Q-M., Coffman, D., Wei., Y-M., 2018. Social cost of carbon under shared socioeconomic pathways. *Glob Env Change* 53, 225-232.
<https://doi.org/10.1016/j.gloenvcha.2018.10.001>

REVIEWER COMMENTS

Reviewer #1 (Remarks to the Author):

The authors have thoroughly revised the manuscript. I particularly appreciate the detailed and well-structured point-by-point response. Overall I believe that this study can make an important contribution to understanding the effects of climate change.

The authors have considerably improved the valuation of the climate regulation service. I am still not convinced concerning the valuation of "habitat". From what I understand, the authors sum up values from a list of different values from the Ecosystem Service Valuation Data bank, while it is not transparent which exact services the authors selected, from which regions, and with which methods they have been valued. This might cause severe problems concerning the aggregation of values. I am aware that many studies based on benefit transfer functions use a similar approach, but they are methodologically highly debated. This should at least be discussed more intensively and the used estimates need to be reported in a more transparent way.

The authors should also clarify how these values differ between the biomes or whether they are constant between the biomes (and change in biomes). I tried to figure this out from the methods and supplementary tables. From what I understand the authors used a constant value for all biomes. If this is really the case then the authors should clarify what we can learn from multiplying this change with a factor, compared to reporting the biome stability itself.

I have serious doubts concerning the soundness of such a valuation. If there is no significant change in the conclusion, except for being able to give an arbitrarily high monetary value, it would be scientifically more sound to stick to the original information and report biome stability as an indicator on the one hand and carbon regulation on the other hand. The authors could still identify relevant hot spots, which I strongly appreciate in the study.

Reviewer #2 (Remarks to the Author):

Although the authors did an effort to improve the methods section based on the suggestions of both reviewers, I still have important concerns on the economic valuation conducted.

It is not clear exactly how opportunity costs (which is in terms of individual benefits) were mixed with social cost of carbon and habitat provision (social benefits).

Opportunity costs are used to estimate the value of economic activities that was forgone for conservation/restoration (e.g. forgone revenues due to agriculture). If climate change affect a particular ecosystem, then the forgone value is in terms of the ecosystem services lost.

My concern regarding how the habitat service value was estimated remains. It seems that the authors are considering this service as a bundle of services, where in reality this is a individual supporting service (sometimes considered a function instead of a service).

Point-by-point response to reviewer's comments

1) Definition of habitat service

R2: *“My concern regarding how the habitat service value was estimated remains. It seems that the authors are considering this service as a bundle of services, where in reality this is a individual supporting service (sometimes considered a function instead of a service).”*

Our definition of habitat as a service is based on a remark by reviewer 1 from the last revision round. Therein, the reviewer stated: *“Concerning the valuation of habitat shifts, the TEEB states that habitat services are valued through other ES i.e. following the CICES cascade.”*

More specifically, the original TEEB document (Pascual et al. 2010) approaches the valuation through a total economic value framework. There, habitat services are described as a group of supporting services (including e.g., primary production, nutrient cycling and soil formation), which form the basis for other categories of services and thus should be valued through their value. A similar categorization is made in Costanza et al. (2017), who group habitat together with regulating and supporting services. Following this, our approach collected values from a large selection of nearly all categories of ecosystem services. Since we did not apply forest management in our simulations we only selected non-provisioning services from the Ecosystem Service Valuation Database (ESVD) and also excluded climate regulation (which we value separately). Altogether, the here applied perspective emphasizes the fundamental supporting value that habitat has for maintaining basic ecosystem functions that enable the provisioning of other ecosystem services along the cascade. To clarify our understanding of habitat services from the beginning, we added the following to the introduction (lines 92-94):

“For this analysis, we consider habitat as a supporting service and value it through the bundled benefits of other ES that result from it as part of the ES cascade (Pascual et al. 2010, Costanza et al. 2017).”

At the same time, it needs to be noted that data availability strongly limits the valuation options for habitat. Even though the ESVD represents the currently largest available collection for ES valuation, it comprises no separate valuations for “habitat”. Since valuations are still generally scarce for our study region, the consideration of multiple ES (including from other tropical regions) was also necessary to arrive at a satisfactory sample size. Therefore, also from a data perspective a bundled valuation represents the best alternative for our case.

Nevertheless, we are aware, that definitions for services related to habitat are highly debated in literature. This is based on the difficulty of determining whether an ecosystem service is “final”, i.e. whether a distinction is made between functions that underpin services (often called “supporting services”) and services which can be related to an economic value in a more direct way (Potschin-Young et al. 2017). The here applied perspective using biome stability to derive potential economic impacts along the ecosystem service cascade thus represents only one of several approaches to describe such benefits to humans. To stress this point, we added the following paragraph to the discussion (lines 327-332):

“The here applied perspective emphasizes the importance of stable habitat (approached through biome stability) for supporting the provisioning of a bundle of other services along the ecosystem service cascade (Pascual et al. 2010, Spangenberg et al. 2014). It needs to be noted, however, that the placement of “habitat” in the context of ecosystem services is still debated because of its complex relations to other services (including classifications as an “ecosystem function” or “intermediate service”, Potschin-Young et al. 2017).”

2) Valuation of habitat service

R1: "I am still not convinced concerning the valuation of "habitat". From what I understand, the authors sum up values from a list of different values from the Ecosystem Service Valuation Data bank, while it is not transparent which exact services the authors selected, from which regions, and with which methods they have been valued. This might cause severe problems concerning the aggregation of values. I am aware that many studies based on benefit transfer functions use a similar approach, but they are methodologically highly debated. This should at least be discussed more intensively and the used estimates need to be reported in a more transparent way."

We performed several filtering and cleaning operations prior to aggregating values from the ESVD (see lines 504-523). To clarify which filters were used and what the final dataset was composed of – especially regarding regions, biomes, ecosystem services and methods – we added this information to the Supplementary (page 20-21, Table S7) and refer to it in the main text (lines 515-517).

Indeed we agree that the aggregation process can have a large influence on the final estimates, since the variance across economic values is often large between different ecosystem services and study contexts. For this reason, we excluded values outside the 5-95% quantile to reduce the influence of outliers and computed average estimates for each biome separately before aggregating them to a final value to avoid the strong bias towards tropical rain forest ecosystems. We are aware, that the such obtained estimates are still subject to considerable uncertainty. However, as stated above, the availability of data matching our study context and the level of harmonization between studies is still very poor. While this stresses the urgent need for more original valuation studies in Central America, conducting such an analysis unfortunately was out of scope for this study. Data from the ESVD certainly need to be used with caution, but at least are readily available and standardized to common units. Therefore, based on currently available data and valuation techniques, up to now benefit transfer represents the method of choice for our study setting. To better explain our rationale, we added the following to the main text (lines 497-503):

"For the valuation of habitat services, we used a benefit transfer approach. While this approach is not without its caveats (Richardson et al. 2015), it represented the most suitable option for our case due to a lack of primary studies in our study region and a lack of resources to conduct our own analysis. To reduce uncertainties and biases related to specific study settings, we aimed to include a broad database and arrive at a central estimate irrespective of the underlying ecosystem, which is in line with our biodiversity conservation perspective and preferable to other strategies of unit value transfer (Richardson et al. 2015)."

3) Biome-dependency of habitat service

R1: "The authors should also clarify how these values differ between the biomes or whether they are constant between the biomes (and change in biomes). I tried to figure this out from the methods and supplementary tables. From what I understand the authors used a constant value for all biomes. If this is really the case then the authors should clarify what we can learn from multiplying this change with a factor, compared to reporting the biome stability itself."

Since we based our economic valuation upon biome stability – and not the biome type itself – we indeed applied a constant value for the whole study region. While a valuation by biome could provide another interesting perspective, there are also several major issues with such an approach. Firstly, a valuation by biomes would multiply existing uncertainties about the economic values, particularly in view of scarce data for some of the biomes. Secondly, biome-dependent values would result in a "ranking" from least to most valuable biomes and allow for gains of habitat value. From a biodiversity conservation perspective, however, this would not be an appropriate valuation approach, since biome shifts represent a destabilization of ecosystem function and could result in

the local extinction of species. Particularly in the fragmented forest landscapes of Central America, species migration may be strongly restricted and alternative habitat may not be available. We added a clarification in the text (lines 500-505):

“To reduce uncertainties and biases related to specific study settings, we aimed to include a broad database and approximate a central estimate irrespective of the underlying ecosystem. This is also in line with our biodiversity conservation perspective and preferable to other strategies of unit value transfer (Richardson et al. 2015). Also, this approach avoids the pitfalls of assigning values by habitat type, which may result in a habitat “ranking” and economic implications that are counter-productive for the conservation of species.”

Therefore, we argue that the application of constant values across all biomes provides a less biased estimation and better matches the biome stability indicator. Why we can still learn considerably from such an economic translation of ecological indicators is explained in the next point.

4) Importance of economic valuation

R1: “I have serious doubts concerning the soundness of such a valuation. If there is no significant change in the conclusion, except for being able to give an arbitrarily high monetary value, it would be scientifically more sound to stick to the original information and report biome stability as an indicator on the one hand and carbon regulation on the other hand.”

Although our valuation method is straightforward, there are still strong arguments for providing monetary values alongside the ecological outputs.

Firstly, ecological indicators are hard to interpret for decision-makers particularly regarding implications for practice. The translation to monetary values on the other hand provides outputs in a known reference system – money – and enables a comparison with other economic data such as the GDP. Considering that Central America includes several low income economies (like Nicaragua, Honduras, Guatemala), a comparison of ecosystem service values with national GDP gives relevant insights into the economic importance and potential of the services. Since we only made this comparison for the whole region so far, we now extended our analysis to show results by country and compare against the respective national GDPs as reference. The thus obtained results highlight the economic relevance of the projected ES losses (up to three times the national GDP) and reveal inequalities in relative economic impacts, which could be severe for the low income countries Belize, Nicaragua and Honduras (described in lines 159-162, 305-307 and the abstract lines 29-31). To better visualize and synthesize these findings with our previous analysis, we added a new figure (Figure 2) and moved the corresponding summary tables to the supplement (Table S2 and Table S3).

Overall, the economic results presented in our study may contribute to raise awareness for the ES values at stake and could contribute to the establishment of payment schemes that may provide monetary incentives to maintain these currently “free” services. Finally, the economic valuation also allows for the consideration of societal preferences, for example regarding time preferences and inter-generational aspects. As such, the application of discounting led to results that were largely different from the patterns of the ecological indicators. Similarly, the choice of price levels had a strong impact on the distribution of economic hotspots, which thus form an important extension to the previously identified ecological hotspots.

To highlight the relevance of the economic part of the study, we added a paragraph to the introduction (lines 68-74):

“The translation of ecological outcomes into monetary units is however of high importance to provide results for decision-makers in an accessible way, raise awareness for neglected economic implications of ES losses and put their relevance into perspective with other economic figures such as the GDP. Specifically in view of a growing number of environmental challenges that decision-makers are faced with, the demand for economic estimates of ecosystem service provision remains high, whereas studies providing guiding information are still scarce (Richardson et al. 2015).”

5) Opportunity costs

R2: “It is not clear exactly how opportunity costs (which is in terms of individual benefits) were mixed with social cost of carbon and habitat provision (social benefits). Opportunity costs are used to estimate the value of economic activities that was forgone for conservation/restoration (e.g. forgone revenues due to agriculture). If climate change affect a particular ecosystem, then the forgone value is in terms of the ecosystem services lost.”

We agree that the term “opportunity costs” may be misleading in the context of climate-induced impacts on ecosystem services. Indeed, opportunity costs are usually associated with benefits that could have been obtained through alternative land uses and are thus often applied in a management context. Since we value ecosystem services as social benefits (as mentioned by reviewer 2), any economic changes are best simply described as gains or losses of ecosystem services (and associated costs to society). Therefore, we changed the terminology by avoiding the term “opportunity costs” and replaced it with “ecosystem service losses” or “costs to society” throughout the paper.

References

Costanza, R., de Groot, R., Braat, L., Kubiszewski, Fioramonti, L., Sutton, P., Farber, S., Grasso, M. (2017) Twenty years of ecosystem services: How far have we come and how far do we still need to go? *Ecosystem Services* 28, 1-6. <https://doi.org/10.1016/j.ecoser.2017.09.008>

Pascual, U., Muradian, R., Brander, L., Gómez-Baggethun, E., Martín-López, B., Verma, M., Armswort, P., Christie, M., Cornelissen, H., Eppink, F., Farley, J., Loomis, J., Pearson, L., Perrings, C., Polasky, S. (2010) The economics of valuing ecosystem services and biodiversity. In: *The economics of ecosystems and biodiversity: Ecological and economic foundations*, 183-256. <https://doi.org/10.4324/9781849775489>

Potschin-Young, M., Czucz, B., Lique, C., Maes, J., Rusch, G.M., Haines-Young, R. (2017) Intermediate ecosystem services: An empty concept? *Ecosystem Services* 27, 124-126. <https://doi.org/10.1016/j.ecoser.2017.09.001>

REVIEWERS' COMMENTS

Reviewer #1 (Remarks to the Author):

The authors now thoroughly discuss the weaknesses of their approach. Therefore I believe it is now scientifically sound and may be offered to the scientific community

Reviewer #2 (Remarks to the Author):

The methods and results are now more robust due to the changes in economic valuation methods. I hope these results can better communicate the cost from climate change on the wellbeing of people and the rest of nature for better decision making in the region.